# Trends in hepatocellular carcinoma and viral hepatitis treatment in older Americans

Joy Jiang[1], Meredith S. Shiels[1], Donna Rivera[2], Marc G. Ghany[3], Eric A. Engels[1], Thomas R. O'Brien[1] *

1 Infections and Immunoepidemiology Branch, Division of Cancer Epidemiology and Genetics, National Cancer Institute, Bethesda, Maryland, United States of America, 2 Division of Cancer Control and Population Sciences, National Cancer Institute, Rockville, Maryland, United States of America, 3 Clinical Research Section, Liver Diseases Branch, National Institute of Diabetes and Digestive and Kidney Diseases, Bethesda, Maryland, United States of America

* obrient@mail.nih.gov

**Data Availability Statement:** The data used in this analysis may be requested from the National Cancer Institute, Surveillance, Epidemiology, and

## Abstract

### Background

Incidence of hepatocellular carcinoma (HCC) had been increasing steadily among older Americans but plateaued in 2015–2017. Chronic infection with hepatitis B virus (HBV) or hepatitis C virus (HCV) are important causes of HCC. The impact of improved treatments for these infections on recent trends in HCC incidence is unclear.

### Aims

To examine the relationship between use of antiviral therapy for chronic viral hepatis and HCC incidence in older Americans.

### Methods

We used 2007–2017 data from the Surveillance, Epidemiology, and End Results—Medicare database to estimate age-standardized incidence rates and average annual percent changes (AAPCs) for viral hepatitis-attributable HCC among individuals ≥66 years. We analyzed data from Medicare Part D to determine the frequency of HBV and HCV treatment utilization in this population.

### Results

Overall HCC incidence increased 10.5%, from 22.2/100,000 in 2007 to 24.5/100,000 in 2017 (AAPC, 1.3%). During that time, HBV-attributable HCC rates decreased from 2.5 to 2.0/100,000 (AAPC, -1.6%), while HCV-attributable HCC rose from 6.6 to 8.0/100,000 (AAPC, 2.0%). HBV treatment among patients with HBV infection increased by 66% (2007, 7.4%; 2015, 12.3%). Treatment for HCV was stable at <2% during 2006–2013 but rose to 6.9% in 2014 and 12.7% in 2015, coinciding with the introduction of direct acting antiviral agents for HCV.

End Results Program https://healthcaredelivery.cancer.gov/seermedicare/obtain/requests.html.

**Funding:** The author(s) received no specific funding for this work.

**Competing interests:** The authors have declared that no competing interests exist.

## Conclusions

A decreased incidence of HBV-attributable HCC corresponded with an increased uptake in treatment for that infection. Despite a marked increase in the effectiveness and frequency of HCV treatment in 2014 and 2015, HCV-attributable HCC had not begun to fall as of 2017.

## Introduction

In the United States, the overall incidence of hepatocellular carcinoma (HCC) increased over several decades, but began to decrease in 2014 [1, 2]. Infection with either hepatitis B virus (HBV) or hepatitis C virus (HCV) is an important driver of HCC, as chronic infection with either virus increases the risk of HCC ~20-fold [3, 4]. Improvements in treatments for chronic hepatitis B (CHB) and chronic hepatitis C (CHC) might explain, at least in part, the recent decline in HCC incidence.

CHB is defined by the persistence of hepatitis B surface antigen (HBsAg) in serum for more than 6 months. Current HBV therapies are not curative, but can lead to the loss of hepatitis B e antigen (HBeAg) and sustained HBV DNA suppression [5], which results in less inflammation and fibrosis progression and a lower risk of developing HCC. HBV treatment originated with the use of interferon-α (IFNα), which was replaced by pegylated-IFNα in 2005. The era of oral therapy for CHB began in 1998 with the approval of lamivudine (S1 Table), which was subsequently supplanted by more effective agents with better resistance profiles [6], including entecavir (2005), tenofovir disoproxil fumarate (2008) and tenofovir alafenamide (2016) [6–10]. Use of these antiviral therapies can reduce the risk of HBV-associated HCC by half [11].

Treatment for CHC differs from that for CHB in that persistent viral elimination, known as a sustained virologic response (SVR), is possible. Treatment of HCV with pegylated-IFNα plus ribavirin yields SVR rates of ~50% [12–14], with lower response rates in patients with advanced fibrosis or cirrhosis, who are at the highest risk of HCC. Both components of that regimen can produce serious adverse effects that further limit its usefulness. The introduction of direct-acting antiviral agents (DAAs), which target HCV proteins, transformed the therapeutic landscape for CHC (S2 Table). In October 2014, the safe, all oral, highly effective fixed dose combination of sofosbuvir/ledipasvir was approved for treatment of CHC, yielding SVR rates >90% [15] followed by sofosbuvir/velpatasvir (2016) and glecaprevir/pibrentasvir (2017). Patients who achieve SVR have a ~70% lower risk for HCC compared to non-responders [16, 17].

There can be a lag between the introduction of an effective treatment and a decreased incidence of disease; therefore, it is important to evaluate trends in HCC incidence in the context of HBV and HCV treatment utilization. Older Americans are a key population in this regard, as the incidence of HCC is highest in those 65 years or older [2] and the prevalence of HCV infection is increased among the cohort born in 1945–1964, which is known as the 'baby boom' generation [18]. The Surveillance, Epidemiology, and End Results (SEER)-Medicare database is a useful resource for assessing the effect of improvements in therapy on the incidence of HBV-attributable and HCV-attributable HCC in older Americans. The linked SEER-Medicare database integrates information on HCC incidence from population-based cancer registries with data from Medicare recipients that includes clinical and treatment data. Data are also available from Medicare Part D, an optional prescription drug benefit that went into effect in 2006.

Previously, we found that HCC rates in Americans aged ≥66 years increased rapidly between 2001 and 2013, partially due to increasing incidence of HBV-attributable and HCV-attributable HCC [19]. Now we have examined rates of HCC attributable to viral hepatitis infections from 2007, the year after Medicare Part D became available, through 2017. To assess the potential effect of changes in viral hepatitis treatment on the incidence of HCC, we examined the proportion of patients with HBV or HCV who were treated for those infections over time and the temporal relationship between changes in the frequency of those treatments with the incidence of hepatitis virus-associated HCC.

## Materials and methods

### Study population

The SEER program collects data from National Cancer Institute (NCI) funded population-based cancer registries. Each registry routinely collects data on all newly diagnosed cancers, including patient demographics, cancer site (International Classification of Diseases for Oncology, Third Edition [ICD-O-3] codes), date of diagnosis and survival. Medicare, a federally funded insurance program for eligible individuals who are 65 years of age and older, provides administrative claims data on medical diagnoses and procedures, including diagnoses of viral hepatitis. We utilized linked SEER-Medicare data from 2007–2017 to identify HCC cases among individuals ≥66 years. For each calendar year, SEER-Medicare selects a 5% random sample of Medicare recipients. The present analysis is based on data for 100% of HCC cases and the 5% sample of the overall population. All analyses were limited to enrollees who: were 66 to 99 years of age; had at least 12 months of both Part A and Part B Medicare coverage outside of a health maintenance organization; had at least 1 Medicare claim of any kind; and had Medicare Part D coverage.

The data for this analysis were accessed for research purposes on September 5, 2023. Because the data for this study were not collected specifically for our study and no one on the study team has access to the subject identifiers linked to the data, this study is not considered human subjects research by the NIH IRB.

### Definitions of HCC, HBV infection, and HCV infection

We defined HCC based on ICD-O-3 site C220, restricted to histology codes 8170–8175. For the years 2007–2015, other diagnoses were defined by International Classification of Diseases, Ninth Revision (ICD-9) codes. For 2016–2017, those diagnoses were based on (ICD-10 codes. For HBV infection, the codes were ICD-9: 070.2, 070.20, 070.21, 070.22, 070.23, 070.3, 070.30, 070.31, 070.32, 070.33, 070.42, 070.52, V02.61; ICD-10: B16, B16.0, B16.1, B16.2, B16.9, B17.0, B18.0, B18.1, B19.1, B19.10, B19.11. HCV infection status was defined using the following codes ICD-9: 070.41, 070.44, 070.51, 070.54, 070.7, 070.70, 070.71, or V02.62; ICD-10: B17.1, B17.10, B17.11, B18.2, B19.2, B19.20, B19.21, Z22.52.

To be classified as infected with HBV or HCV we required either one inpatient claim or two outpatient claims filed at least 30 days apart, restricted to the time frame from 60 months before to 12 months after cancer diagnosis. Individuals with HCC who failed to meet the definition for either HBV or HCV were considered to have HCC that was unrelated to either virus (i.e., not of viral etiology).

We defined other potential etiologic factors for HCC based on the following codes ICD-9 and ICD-10 codes: diabetes mellitus—ICD-9: 250.0–250.9; ICD-10; E08-E13; alcohol-related liver disorders—ICD-9: 571.0, 571.1, 571.2, 571.3, and V11.3, 571.6 in the presence of 291, 303, or 305.0; ICD-10: K70, K70.0, K70.1, K70.10, K70.11, K70.2, K70.3, K70.30, K70.31, K70.4, K70.40, K70.41, K70.9; rare genetic disorders—ICD-9: 270.2 (tyrosinemia); 273.4 (alpha-1

antitrypsin deficiency); 275.0 (hemochromatosis); 275.1 (Wilson disease); 277.1 (porphyria) ICD-10: E70.21 (tyrosinemia); E88.01 (alpha-1 antitrypsin deficiency); E83.11, E83.110, E83.111, E83.118, E83.119 (hemochromatosis); E83.01 (Wilson disease); E80.0 (hereditary erythropoietic porphyria).

## Prevalence of hepatitis virus infection and use of medications

To ascertain the prevalence of infection with HBV or HCV for each calendar year, we divided the number of recipients in the 5% sample who had a diagnosis of HBV or HCV infection by the total number of recipients in that year's sample. Individuals with HCC were included in this calculation only if they were included in the 5% random sample of Medicare recipients with a cancer diagnosis. Once a recipient was determined to have HBV or HCV infection that diagnosis was carried forward for all subsequent years. To calculate the proportion of patients with HBV or HCV who received treatment for that infection each year, we divided the number of patients with a Medicare Part D claim for treatment of HBV (S1 Table) or HCV (S2 Table) by the number of patients with the corresponding infection. Interferon-α-based therapies can be used to treat either HBV or HCV; therefore, these medications were considered as a treatment for both viruses.

## Statistical analysis

We determined the frequency of HCC, overall and by hepatitis virus status, stratifying by demographic characteristics and the presence of certain HCC risk factors. To calculate overall annual HCC incidence, we divided the number of HCC cases in a given calendar year by 20 times the number of individuals included in the 5% Medicare sample for that year. In calculating the annual incidence of HBV-associated and HCV-associated HCC, we used the number of HCC cases associated with that virus as the numerator. We age-standardized these incidence rates to the 2000 US population. We calculated the average annual percent change (AAPC) in HCC incidence (overall, HBV-attributable, HCV-attributable and HBV/HCV-unrelated) by age group, sex and race/ethnicity for the period 2007–2017. We used Joinpoint 4.7.0.0 [20] (NCI, Bethesda, MD) to ascertain statistically significant changes in HCC incidence over time and to estimate the annual percent change (APC) for such periods.

We used SEER*Stat 8.3.6 [21] (NCI, Bethesda, MD) to estimate age-standardized incidence rates of HBV- and HCV-attributable HCC, and SAS 9.4 (SAS Institute Inc., Cary, NC) to analyze data on treatment from Medicare Part D.

Use of SEER-Medicare data is subject to the agreement that no findings from the data may be released if such findings contain data elements that might allow patient identification. Specifically, frequencies of <11 and percentages based on frequencies <11 must be suppressed. For that reason, such results are not presented in this report.

## Results

### Characteristics of individuals with HCC

There were 19,552 cases of HCC among Medicare beneficiaries between 2007 and 2017 (Table 1). Of these, 62.9% of cases could not be attributed to infection with a hepatitis virus, 9.8% were HBV-related and 32.1% were HCV-related (including 4.8% that were jointly attributable to HBV and HCV). Men contributed about two-thirds of both viral hepatitis-related and viral hepatitis-unrelated HCC cases. HBV-attributable and HCV-attributable HCC cases had similar age distributions, with most occurring in the 66-75-year age group. In contrast, those with HBV/HCV-unrelated HCC tended to be older with the majority over 75 years of

**Table 1. Characteristics of individuals with hepatocellular carcinoma, by hepatitis B virus (HBV) and hepatitis C virus (HCV) infection status, SEER-Medicare, 2007–2017.**

| | Total | | HBV+ | | HCV+ | | HBV/ HCV-Unrelated | |
|---|---|---|---|---|---|---|---|---|
| | Count | % | Count | % | Count | % | Count | % |
| **Total** | 19,552 | | 1,917 | 9.8% | 6,267 | 32.1% | 12,294 | 62.9% |
| **Sex** | | | | | | | | |
| Male | 13,526 | 69.2% | 1,358 | 70.8% | 4,112 | 65.6% | 8,701 | 70.8% |
| Female | 6,026 | 30.8% | 559 | 29.2% | 2,115 | 33.8% | 3,593 | 29.2% |
| **Age group, y** | | | | | | | | |
| 66–75 | 10,687 | 54.7% | 1,217 | 63.5% | 4,268 | 68.1% | 5,815 | 47.3% |
| 76–85 | 6,675 | 34.1% | 561 | 29.3% | 1,634 | 26.1% | 4,735 | 38.5% |
| 86+ | 2,190 | 11.2% | 139 | 7.3% | 364 | 5.8% | 1,744 | 14.2% |
| **Race/ethnicity** | | | | | | | | |
| White | 13,906 | 71.1% | 588 | 30.7% | 3,635 | 58.0% | 10,035 | 81.6% |
| Black | 1,805 | 9.2% | 182 | 9.5% | 1,186 | 18.9% | 570 | 4.6% |
| Asian | 1,844 | 9.4% | 796 | 41.5% | 760 | 12.1% | 601 | 4.9% |
| Hispanic | 808 | 4.1% | 59 | 3.1% | 301 | 4.8% | 488 | 4.0% |
| Other/unknown | 1,189 | 6.1% | 292 | 15.2% | 385 | 6.1% | 600 | 4.9% |
| **Comorbidities** | | | | | | | | |
| Alcohol-related liver disorders | 3,947 | 20.2% | 293 | 15.3% | 1,391 | 22.2% | 2,445 | 19.9% |
| Diabetes mellitus | 11,664 | 59.7% | 1,047 | 54.6% | 3,250 | 51.9% | 7,895 | 64.2% |
| Rare genetic disorders | 596 | 3.1% | 35 | 1.8% | 153 | 2.4% | 432 | 6.9% |
| Obesity | 3,371 | 17.2% | 179 | 9.3% | 735 | 11.7% | 2,551 | 20.8% |

age. The racial/ethnic distribution of HCC cases differed by etiology. Asian individuals accounted for the highest proportion of HBV-attributable HCC cases (41.5%), while non-Hispanic White individuals accounted for most HCV-attributable (58.0%) and HBV/HCV-unrelated (81.6%) cases. Diabetes mellitus, which was reported in 59.7% of HCC cases overall, was similarly present in about half of those with HBV-attributable HCC or with HCV-attributable HCC, as well as in 64.2% of those without evidence of viral hepatitis. Alcohol-related liver disorders were more prevalent among HCV-attributable cases (22.2%) than those without evidence of viral hepatitis infection (19.9%). HCC-associated genetic disorders were reported in 6.9% of HBV/HCV-unrelated HCC cases and ~2% of those with viral hepatitis.

## Trends in HCC incidence

Age-standardized rates of HCC rose 1.3% per year overall from 2007 (22.2 per 100,000) to 2017 (24.5 per 100,000; Table 2). Men had a significant increase in overall HCC incidence during 2007–2017 (AAPC, 1.4%), while women had a smaller, statistically non-significant increase (AAPC, 0.2%). An examination of HCC incidence by age showed a significant increase of 1.7% per year among those 66–75 years that drove the change in HCC incidence. We observed markedly different trends in HCC rates by race/ethnicity. Significant increases were present among both non-Hispanic White (AAPC, 1.9%) and non-Hispanic Black (AAPC, 3.4%) individuals. In contrast, HCC incidence decreased significantly in Asian individuals (AAPC, -2.8%) and individuals of unknown or other race/ethnicity (AAPC, -4.2%). The upturn in the HCC rate was driven by statistically significant increases in the incidence of HCV-attributable HCC (2.0% per year) and HBV/HCV-unrelated HCC (1.1% per year). The incidence of HBV-attributable HCC fell by 1.6% per year during this period (Fig 1).

**Table 2. Average annual percent change (AAPC) with 95% confidence interval (CI) in age standardized hepatocellular carcinoma incidence rates per 100,000, overall and by hepatitis B virus (HBV) and hepatitis C virus (HCV) infection status, SEER-Medicare, 2007–2017.**

| Demographic | All HCC Cases | | | HBV-Attributable HCC Cases | | | HCV-Attributable HCC Cases | | | HBV/HCV-Unrelated HCC Cases | | |
|---|---|---|---|---|---|---|---|---|---|---|---|---|
| | 2007 Rate | 2017 Rate | AAPC (95% CI) [†] | 2007 Rate | 2017 Rate | AAPC (95% CI) [†] | 2007 Rate | 2017 Rate | AAPC (95% CI) [†] | 2007 Rate | 2017 Rate | AAPC (95% CI) [†] |
| **Total** | 22.2 | 24.5 | **1.3 (0.7 to 1.9)** | 2.5 | 2.0 | -1.6 (-3.5 to 0.3) | 6.6 | 8.0 | **2.0 (0.8 to 3.2)** | 14.3 | 15.4 | **1.1 (0.4 to 1.8)** |
| **Sex** | | | | | | | | | | | | |
| **Male** | 35.8 | 41.6 | **1.4 (0.8 to 2.1)** | 3.9 | 3.6 | -1.0 (-3.1 to 1.1) | 9.6 | 13.6 | **3.3 (2.2 to 4.4)** | 24.0 | 25.9 | 0.8 (-0.1 to 1.8) |
| **Female** | 12.5 | 11.5 | 0.2 (-1.2 to 1.6) | 1.4 | 0.72 | -3.4 (-6.4 to 0.4) | 4.3 | 3.5 | -0.8 (-3.8 to 2.2) | 7.7 | 7.6 | 0.8 (-0.2 to 1.9) |
| **Age group, y** | | | | | | | | | | | | |
| **66–75** | 22.8 | 25.1 | **1.7 (1.1 to 2.3)** | 3.0 | 2.3 | -1.9 (-3.6 to 0.0) | 7.5 | 11.1 | **3.9 (2.5 to 5.4)** | 13.6 | 12.8 | 0.6 (-0.6 to 1.8) |
| **76–85** | 24.2 | 26.4 | 0.6 (-0.7 to 2.0) | 2.4 | 1.8 | -1.9 (-5.7 to 2.1) | 6.7 | 5.0 | -1.9 (-5.5 to 1.9) | 16.5 | 20.2 | **1.5 (0.6 to 2.4)** |
| **86+** | 14.2 | 17.3 | 1.8 (0.8 to 2.7) | 0.6 | 1.3 | 2.8 (-2.7 to 8.6) | 2.3 | 2.9 | 0.9 (-1.7 to 3.7) | 11.7 | 13.6 | **1.8 (0.7 to 2.9)** |
| **Race/ethnicity** | | | | | | | | | | | | |
| **White** | 18.6 | 21.6 | **1.9 (1.2 to 2.6)** | 0.8 | 0.9 | 0.3 (-2.1 to 2.7) | 4.3 | 6.1 | **3.7 (2.5 to 4.9)** | 14.0 | 15.3 | **1.3 (0.4 to 2.3)** |
| **Black** | 23.1 | 36.0 | **3.4 (1.5 to 5.2)** | 2.2 | 2.9 | 1.8 (-2.4 to 6.2) | 12.2 | 25.0 | 5.3 (3.0 to 7.6) | 10.3 | 10.0 | -0.6 (-3.3 to 2.2) |
| **Asian** | 77.9 | 47.9 | **-2.8 (-4.8 to -0.7)** | 37.8 | 20.4 | **-4.3 (-6.7 to -1.8)** | 32.0 | 15.6 | **-4.8 (-8.2 to -1.2)** | 23.3 | 17.2 | 0 (-3.0 to 3.0) |
| **Hispanic** | 43.3 | 42.6 | 0.4 (-2.1 to 2.9) | 2.8 | 3.1 | -1.4 (-9.8 to 7.8) | 16.9 | 13.6 | -0.5 (-4.2 to 3.3) | 26.4 | 27.6 | 0.8 (-2.7 to 4.5) |
| **Other/ unknown** | 40.3 | 30.7 | **-4.2 (-6.6 to -1.8)** | 10.3 | 5.8 | **-5.6 (-10.5 to -0.5)** | 19.5 | 8.1 | **-7.1 (-10.5 to -3.6)** | 15.8 | 17.8 | -2.7 (-5.7 to 0.4) |

Abbreviations: AAPC, average annual percent change; HBV, hepatitis B virus; HCC, hepatocellular carcinoma; HCV, hepatitis C virus; SEER, Surveillance, Epidemiology, and End Results; 95% CI, 95% confidence interval

[†] - Statistically significant AAPCs are denoted in bold text.

[§] - Statistically significant joinpoints are as follows:

HCV-attributable Female (2007–2013: 4.1 [-0.1 to 8.4]; 2013–2017: -9.0 [-16.0 to -1.5])

76–85 years (2007–2013: 3.6 [-1.3 to 8.8]; 2013–2017: -12.2 [-20.9 to -2.6])

Asian (2007–2014: -0.9% [-5.4 to 3.8]; 2014–2017: -18.9% [-34.6 to 0.6]

HBV/HCV-unrelated Other/unknown race/ethnicity (2007–2009: 23.1% [-11.2 to 70.5]; 2009–2017: -5.1% [-7.7 to -2.2]

Examining the incidence of HBV-attributable HCC in more detail revealed non-significant decreases in the AAPC for both men and women (Table 2; Fig 2A). By age group, there were non-significant declines in the AAPC among those 66–75 and 76–85 years and non-significant increases among those 86+ years. Examining HBV-attributable HCC by race/ethnicity, the AAPC declined significantly among Asian individuals at -4.3% per year and among those of other or unknown race/ethnicity at -5.6% per year with no changes in trend over the period.

The incidence of HCV-attributable HCC increased significantly among men (AAPC, 3.3%), but not among women (AAPC, -0.8%; Table 2; Fig 2B). However, among women, the rate increased non-significantly at 4.1% from 2007 to 2013 and decreased significantly at -9.0% from 2013–2017 (Table 2 footnote). By age group, there was a significant rise in incidence among those 66–75 years of age at 3.9% per year, while rates for other age groups did not change significantly over the 2007–2017 time period. Among those 76–85 years of age, the rate increased non-significantly by 3.6% from 2007 to 2013 and then declined significantly by -12.2% from 2013–2017 (Table 2 footnote). By race/ethnicity, HCV-attributable HCC rates rose significantly among non-Hispanic White individuals (3.7% per year), whereas rates declined significantly among Asian individuals (AAPC, -4.8%) and those of other or unknown race/ethnicity (AAPC, -7.1%). In the joinpoint analysis, the rate for Asian individuals decreased non-significantly (-0.9% per year) from 2007 to 2014 and then decreased significantly at -18.9% per year between 2014 and 2017 (Table 2 footnote).

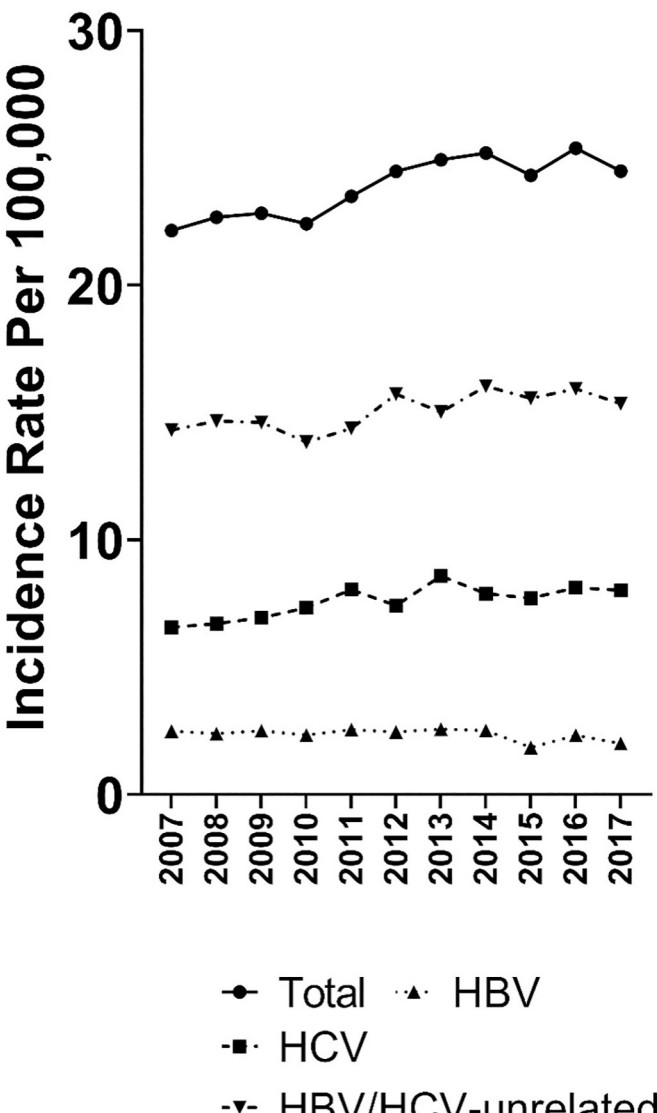

**Fig 1. Age-standardized hepatocellular carcinoma (HCC) incidence rates, by viral hepatitis status, the surveillance, epidemiology, and end results–medicare database, 2007–2017.** Rates of hepatitis B virus (HBV)-attributable HCC and hepatitis C virus (HCV)-attributable HCC are not mutually exclusive. *Statistically significant joinpoint.

HCC that was unrelated to either HBV infection or HCV infection was not the primary focus of this study. However, given that such cases contributed a large proportion of HCC cases in this analysis, among patients without viral hepatitis, we examined the incidence of HCC in relation to certain comorbidities that are known to increase the risk of HCC (S1 Fig). From 2007 to 2017, the incidence of HCC that occurred in individuals who also had either diabetes mellitus or obesity increased with an AAPC of 2.0%. The AAPC for HCC occurring in individuals with alcohol-related liver disorders was 3.8%, while HCC cases occurring among individuals with a rare genetic disorder increased at a more modest rate (AAPC, 1.2%). All of those increases were statistically significant.

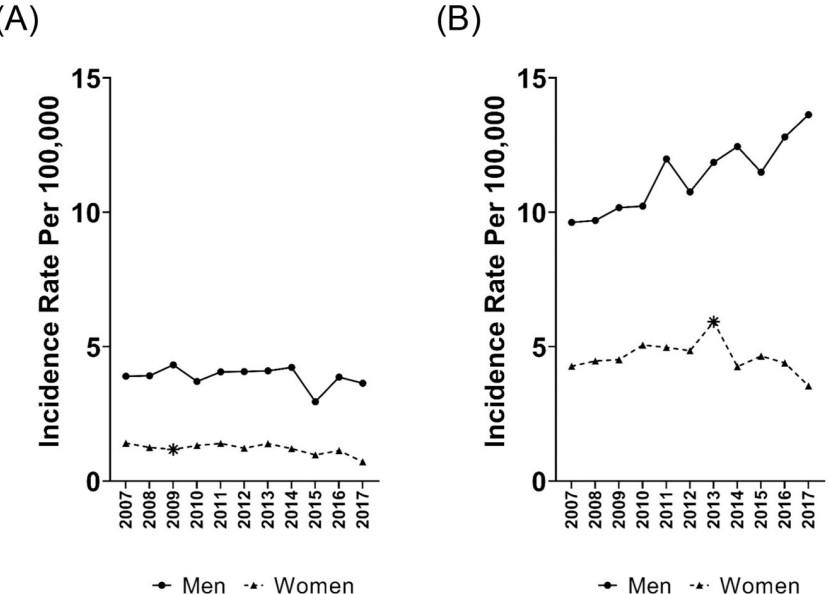

**Fig 2. Age-standardized incidence rates by sex, for (A) hepatitis B virus-attributable hepatocellular carcinoma; (B) hepatitis C virus-attributable hepatocellular carcinoma—Surveillance, Epidemiology, and End Results–Medicare database, 2007–2015.** *Statistically significant joinpoint.

## Trends in viral prevalence and treatment for hepatitis virus infections

In this population of Medicare beneficiaries with Part D coverage, the prevalence of a claim related to HBV infection was 0.53% in 2007, 0.58% in 2008 and then consistently ≥0.62% during the ensuing years (Fig 3A). Among these individuals with evidence of HBV infection, the proportion with a claim for HBV treatment rose from 7.4% in 2007 to 9.8% in 2012 (the year before the decline in HBV-associated HCC began) (Fig 4A). The proportion of patients receiving HBV treatment continued to increase, reaching 12.3% in 2017, while the incidence of HBV-attributable HCC continued to decrease during this period.

The prevalence of a claim related to HCV infection in this population rose from 0.86% in 2007 to 1.28% in 2017 (Fig 3B). Among Medicare beneficiaries with HCV infection, the proportion with a claim for HCV treatment remained below 2% until 2014 when it rose to 6.9% and then to 12.7% in 2015. Treatment decreased to 9.3% in 2016 and 6.5% in 2017. HCC incidence was highest in 2013 (8.58/100,000) and decreased to 7.90 in 2014. Despite the increase in HCV treatment, there was no decrease in the incidence of HCV-associated HCC in the SEER-Medicare population from 2014 to 2017 (Fig 4B).

## Discussion

From 2007 to 2017, overall HCC incidence among Medicare patients rose an average of 1.3% per year for a cumulative increase of 10.5%. That change was driven by rising rates of HCV-attributable HCC (2.0% per year) and HBV/HCV-unrelated HCC (1.1% per year), which were partially offset by a decreasing incidence of HBV-attributable HCC cases (-1.6% per year).

Disaggregating rates by race/ethnicity revealed marked differences in trends in incidence for HBV-attributable HCC. Among Asian individuals, who contributed 41.5% of HBV-attributable HCC cases overall, the incidence of HCC due to HBV decreased 2.8% per year during

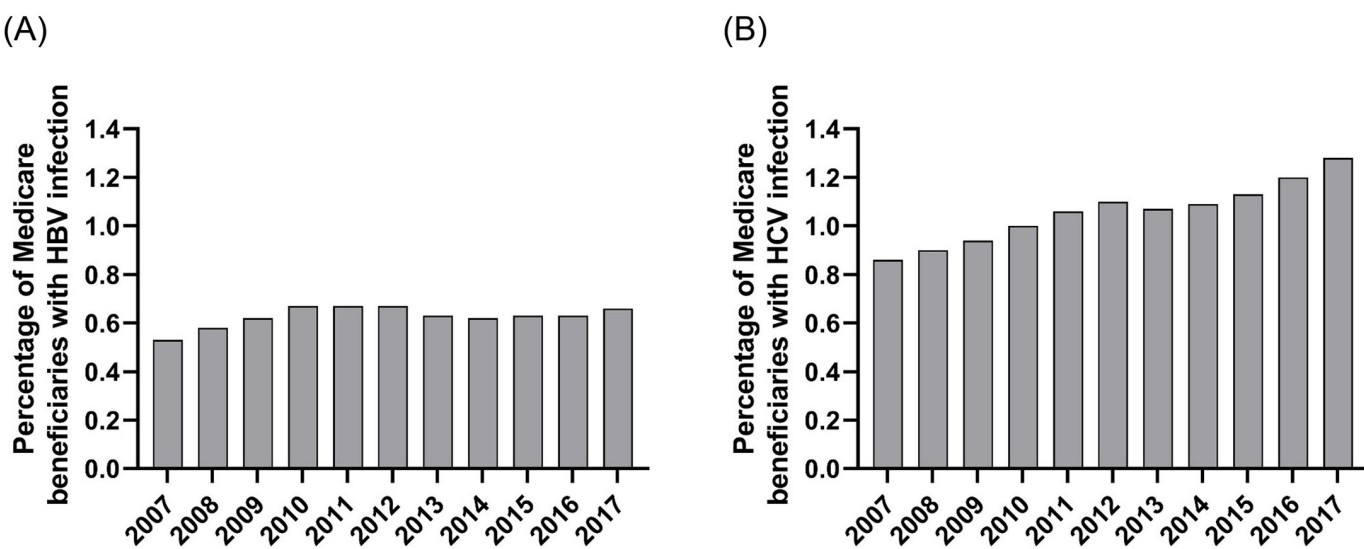

**Fig 3. Prevalence of infection with (A) hepatitis B virus or (B) hepatitis C virus hepatocellular carcinoma among Medicare Part D beneficiaries, 2007–2017.**

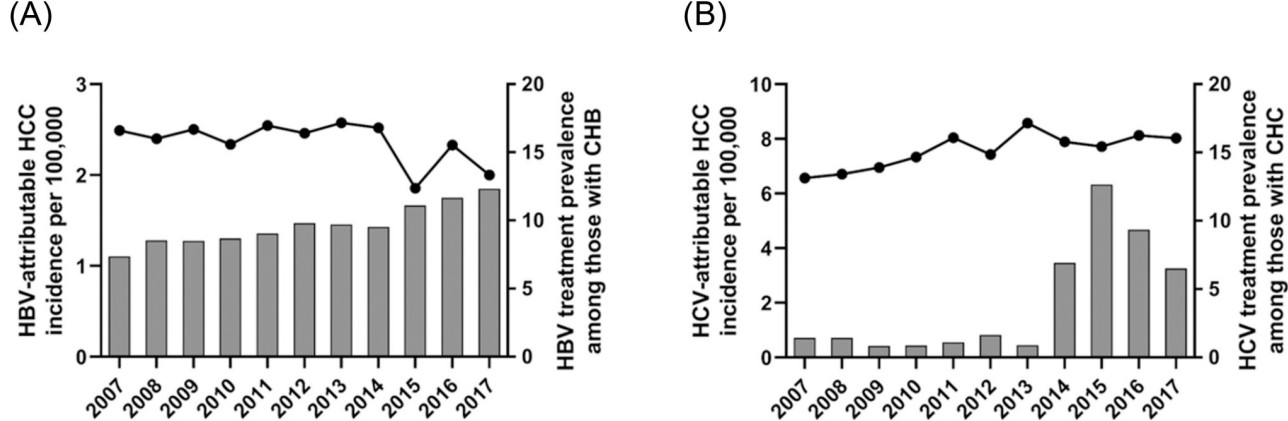

**Fig 4. Incidence of (A) hepatitis B virus (HBV)-attributable hepatocellular carcinoma (HCC) and (B) hepatitis C virus (HCV)-attributable HCC in the SEER-Medicare population and frequency of treatment for the corresponding infection among Medicare Part D beneficiaries, 2007–2017.**

2007–2017. Individuals of other or unknown race/ ethnicity contributed 15.2% of HBV-attributable HCC cases, with incidence decreasing 5.6% per year over the study period in that ill-defined group. Among the remaining racial/ethnic groupings, AAPC for the incidence of HBV-attributable HCC ranged from -1.4% to +1.8% without any statistically significant trends. Therefore, the overall 1.6% decrease in HCC linked to HBV reflects decreases in Asians and individuals of other or unknown race/ ethnicity.

Examining the proportion of patients with HBV infection who were receiving treatment revealed a gradual increase from 7.4% in 2007 to 12.3% in 2017. The availability of more potent treatments for HBV with lower rates of antiviral resistance beginning in ~2006 may have contributed to the decrease in HBV-attributable HCC we observed.

The incidence of HCV-attributable HCC incidence rose between 2007 and 2017. That increase was driven by rising rates among men, individuals 66–75 years of age and White individuals, and it occurred despite a decrease in HCV-attributable HCC rates among Asian individuals and individuals of other or unknown race/ ethnicity. Given that HCC usually develops 30–40 years after initial infection, this overall increase in HCV-associated HCC likely represents the large cohort of persons infected with HCV in the early 1970s who acquired that infection from injection drug use or other high-risk behaviors [18].

Medicare claims for HCV treatment were low during 2007 to 2013, but then increased markedly in 2014 and again in 2015, corresponding with FDA approval of two DAA regimens in late 2014. Treatment guidelines issued in 2015 called for treatment of all people with HCV except those with limited life expectancy due to non—liver-related comorbid conditions [22]. By July 2015 all Part D plans covered at least one of these new, highly effective treatments for CHC, although delays in uptake and treatment may have resulted from relatively high coinsurance or prior authorization requirements [23]. During this period, patients with cirrhosis were prioritized for treatment. Notably, we saw no decrease in the incidence of HCV-attributable HCC through 2017. Data from cohort studies indicate that differences in HCC incidence between people with HCV who achieve SVR and those who remain infected begin to be seen ~6 months after successful treatment. Although those who have been successfully treated are at markedly lower risk of HCC over time, some risk continues for a number of years, especially among those with more advanced liver disease [17, 24]. Due to those factors, insufficient time may have passed between the introduction of highly effective HCV therapies in late 2014 and the end our follow up in 2017 to affect HCC incidence in this population during the study period.

Previously we examined recent trends in the incidence of HCC based on SEER data that were not linked to Medicare records. While we have shown that overall HCC rates are in decline [1, 2], consistent with our findings in this SEER-Medicare population, there is as yet is no evidence of a decrease in HCC incidence among those older than 65 years in that larger dataset [25].

An important strength of our study is that SEER-Medicare provides data from a large sample of population-based data. The information on medical claims contributed by the data from Medicare allowed us to study virus-specific trends in HCC incidence, something that is not feasible using SEER cancer incidence data alone. That allowed us to reveal that the rate of HBV-attributable HCC was decreasing, despite the increasing incidence of HCC overall. Overlaying the information on medication-based treatment obtained from Medicare Part D data allowed us to compare virus specific time trends in HCC incidence to those for anti-viral treatment.

Our study has several important limitations related to the observational data used in the analyses. As Medicare coverage is generally limited to those who are 65 years of age or older, we could not determine if our findings generalize to younger individuals. Medicare Part A and B claims are unavailable for those who obtain Medicare coverage from a health maintenance organization and such people may differ from the general Medicare population. We would have liked to compare the incidence of cirrhosis and HCC between patients with chronic hepatitis B who did or did not receive antiviral therapy. However, SEER-Medicare is based on a series of annual 5% random samples of Medicare recipients, which makes it impossible to define a prospective cohort within that database. Medicare claims data did not allow us to distinguish between chronic and resolved viral hepatitis infections or to determine whether a patient with HBV infection met guidelines for antiviral treatment. Our classification of HCC cases as to viral etiology (and the resulting attributable percentages) are likely reliable because patients with HCC generally undergo testing for HCV and HBV. However, CHB and CHC

can be present without causing serious disease, therefore, these infections were likely under-ascertained in individuals without cancer. Because people with clinically documented infection (i.e., claims) form the denominator for our estimate of the proportion of infected people who received treatment, underestimation of the proportion who are infected would lead to an over-estimation of the proportion of infected patients who received treatment.

For both HBV-associated and HCV-associated HCC, we saw higher incidence rates among Black, Asian and Hispanic individuals than in the White population. The SEER-Medicare database is not well suited for addressing the underlying causes of these racial/ ethnic disparities, as it does not include detailed information pertaining to social determinants of health such as access to medical care or socioeconomic status, however, disparities in obtaining needed treatment and information regarding coverage among minority populations have been previously reported [26, 27]. The structural and social context of the present study's epidemiological observations should be the focus of future work.

In summary, HCC rates overall increased from 2007–2017 among Americans ≥66 years of age, with HCV-attributable HCC and viral hepatitis unrelated HCC contributing substantially to this increase. The changing therapeutic landscape for viral hepatitis should continue to mitigate HCC risk among older Americans. Continued monitoring of HCC trends in relation to viral hepatitis treatment may provide valuable feedback on the uptake and effectiveness of these medications on the population level and could inform policies regarding the value of these therapies in preventing HCC, an important cause of cancer mortality.

## Supporting information

**S1 Fig. Age-standardized hepatocellular carcinoma incidence rates among patients without viral hepatitis, by status for diabetes and/or obesity, alcohol-related liver disorders and rare genetic disorders, the surveillance, epidemiology, and end results–Medicare database, 2007–2017.** The occurrence of risk factors is not mutually exclusive.
(TIF)

**S1 Table. Timeline of FDA approval of HBV treatments.**
(DOCX)

**S2 Table. Timeline of FDA approval of HCV treatments.**
(DOCX)

## Acknowledgments

"This study used the linked SEER-Medicare database. The interpretation and reporting of these data are the sole responsibility of the authors. The authors acknowledge the efforts of the National Cancer Institute; Information Management Services (IMS), Inc.; and the Surveillance, Epidemiology, and End Results (SEER) Program tumor registries in the creation of the SEER-Medicare database. We acknowledge the statistical support of David Castenson and Michael Barrett (IMS Inc)."

## Author Contributions

**Conceptualization:** Joy Jiang, Meredith S. Shiels, Thomas R. O'Brien.

**Formal analysis:** Joy Jiang, Meredith S. Shiels, Thomas R. O'Brien.

**Methodology:** Meredith S. Shiels.

**Supervision:** Thomas R. O'Brien.

**Writing – original draft:** Joy Jiang, Meredith S. Shiels, Thomas R. O'Brien.

**Writing – review & editing:** Joy Jiang, Meredith S. Shiels, Donna Rivera, Marc G. Ghany, Eric A. Engels, Thomas R. O'Brien.

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
