## [Decision Letter · Decision Letter 0]

20 May 2024

PONE-D-24-13784Trends in Hepatocellular Carcinoma and Viral Hepatitis Treatment in Older AmericansPLOS ONE

Dear Dr. O'Brien,

Thank you for submitting your manuscript to PLOS ONE. After careful consideration, we feel that it has merit but does not fully meet PLOS ONE’s publication criteria as it currently stands. Therefore, we invite you to submit a revised version of the manuscript that addresses the points raised during the review process.

Please, consider carefully reviewers´comments and try to update the manuscript appropriately. 

We look forward to receiving your revised manuscript.

Kind regards,

Sona Frankova, M.D., Ph.D. 

Academic Editor

PLOS ONE

2. PLOS requires an ORCID iD for the corresponding author in Editorial Manager on papers submitted after December 6th, 2016. Please ensure that you have an ORCID iD and that it is validated in Editorial Manager. To do this, go to ‘Update my Information’ (in the upper left-hand corner of the main menu), and click on the Fetch/Validate link next to the ORCID field. This will take you to the ORCID site and allow you to create a new iD or authenticate a pre-existing iD in Editorial Manager. Please see the following video for instructions on linking an ORCID iD to your Editorial Manager account: https://www.youtube.com/watch?v=_xcclfuvtxQ.

Reviewers' comments:

Reviewer's Responses to Questions

**Comments to the Author**

1. Is the manuscript technically sound, and do the data support the conclusions?

Reviewer #1: Yes

Reviewer #2: Yes

2. Has the statistical analysis been performed appropriately and rigorously? 

Reviewer #1: Yes

Reviewer #2: Yes

3. Have the authors made all data underlying the findings in their manuscript fully available?

Reviewer #1: Yes

Reviewer #2: Yes

4. Is the manuscript presented in an intelligible fashion and written in standard English?

Reviewer #1: Yes

Reviewer #2: Yes

5. Review Comments to the Author

Reviewer #1: SUMMARY:

Between 2007 and 2017, researchers observed a concerning trend among Medicare beneficiaries aged 66 and older: the overall incidence of hepatocellular carcinoma (HCC) rose by an average of 1.3% per year. This increase appears to be fueled by two key factors: a rise in both hepatitis C virus (HCV)-related HCC and HCC cases unrelated to any identifiable viral hepatitis infection. These growing subsets of HCC cases partially offset a simultaneous decline in hepatitis B virus (HBV)-related HCC incidence. Further, the study unveiled that the decreasing trend in HBV-related HCC coincided with an increase in HBV treatment rates. The proportion of Medicare beneficiaries with HBV infection who received antiviral therapy gradually rose from 7.4% in 2007 to 12.3% in 2017. This suggests that the availability of more potent and effective HBV treatments with lower resistance rates may have played a significant role in reducing HCC risk within this population. In contrast, the incidence of HCV-related HCC increased significantly during the study period, primarily driven by rising rates among men, those aged 66-75, and non-Hispanic White individuals. These trends are likely attributable to a large cohort of individuals who contracted HCV during the 1970s through injection drug use and other high-risk behaviors. While the approval of highly effective direct-acting antivirals (DAAs) in 2014 led to a surge in HCV treatment, the study period may have been too short to observe a corresponding impact on HCC rates at the population level. It typically takes time for the beneficial effects of successful HCV treatment to translate into a decreased risk of HCC, especially among individuals with advanced liver disease. The study also exposed stark racial and ethnic disparities in HCC trends. Notably, rates of HBV-related HCC fell significantly among Asian individuals and those of other/unknown race or ethnicity. In contrast, HCV-related HCC rose significantly in the non-Hispanic White population, while declining in Asians and individuals of other/unknown race or ethnicity. These findings underscore the complexities of HCC epidemiology and highlight the need for further research to understand and address the unequal burden of liver cancer across different racial and ethnic groups. In conclusion, this study offers valuable insights into the shifting landscape of HCC incidence and its intricate relationship with the management of viral hepatitis. While effective HBV treatments appear to be curbing HCC risk linked to this virus, the delayed impact of HCV therapies calls for continued surveillance. More importantly, the marked racial and ethnic disparities bring attention to the necessity of tailored prevention and intervention strategies to reduce the overall burden of HCC.

LIMITATIONS:

Limitations of Medicare Data: The study heavily relies on Medicare claims data, which has inherent limitations. (1) Age restriction: The focus on Medicare beneficiaries (65 and above) limits the generalizability of findings to younger populations who may have different HCC risks and treatment patterns. (2) Missing data: Patients enrolled in Medicare Advantage plans are not represented in the data, introducing a potential bias in the analysis. (3) Incomplete Information: Medicare claims may not accurately capture certain risk factors (e.g., alcohol use, insulin resistance) or distinguish between chronic and resolved viral infections. This could affect the accuracy of attributing HCC cases to specific causes. (4) Short Observation Period: The study's observation window (2007-2017) might not be long enough to fully assess the impact of direct-acting antiviral therapy on HCV-related HCC incidence. Longer follow-up would be needed to see if the surge in HCV treatment translates to a decrease in HCC rates.

Unexplored Disparities: While the study highlights racial/ethnic disparities, it does not delve into potential underlying causes. Factors like socioeconomic status, access to care, genetic predisposition, and differing responses to treatment across racial groups could be explored further or addressed in the discussion.

Scope of "Non-Viral" HCC: The large proportion of HCC cases classified as "unrelated to viral hepatitis" warrants deeper investigation. This category likely includes HCC with various origins (e.g., NAFLD, alcohol-related liver disease, genetic conditions) that might have different trends and risk factors. A more nuanced breakdown of this category, or additional discussion, could provide better insights.

Reviewer #2: Very interesting work demonstrating the trends in incidence of HCC among older Americans, which is rising, attributed mostly to the incidence of hepatitis C and non-viral causes, while the number of cases attributable to HBV infection is declining. It is interesting to see that almost 60% of HCC cases has diabetes, highlighting again the important role of metabolic disturbances typical of the modern lifestyle.

6. PLOS authors have the option to publish the peer review history of their article (what does this mean?). If published, this will include your full peer review and any attached files.

Reviewer #1: No

Reviewer #2: No

---

## [Author Response · Author response to Decision Letter 0]

9 Jul 2024

Reviewer #1 suggested that we investigate HCC cases classified as "unrelated to viral hepatitis." In response to that comment, we performed additional analyses that showed an increase in the incidence of HCC occurred in individuals who also had either diabetes mellitus or obesity (main components of NASH), as well as those with an alcohol-related liver disorders or a rare genetic disorder. We present those new data on pages 7-8 of the manuscript and in Supplementary Figure 1. 

Both reviewers suggested that additional comment on the racial/ethnic disparities in HCC incidence we observed was warranted. To that end, we have added a new paragraph in the discussion (pages 9-10) to address that important point.

---

## [Editor Report · Decision Letter 1]

11 Jul 2024

Trends in Hepatocellular Carcinoma and Viral Hepatitis Treatment in Older Americans

PONE-D-24-13784R1

Dear Dr. O'Brien,

We’re pleased to inform you that your manuscript has been judged scientifically suitable for publication and will be formally accepted for publication once it meets all outstanding technical requirements.

Kind regards,

Sona Frankova, M.D., Ph.D. 

Academic Editor

PLOS ONE

Additional Editor Comments (optional):

Please update the graphs so that the x-axes 1 and 2 (left and right) desciptions have the same direction.

---

## [Editor Report · Acceptance letter]

25 Jul 2024

PONE-D-24-13784R1 

PLOS ONE

Dear Dr. O’Brien, 

I'm pleased to inform you that your manuscript has been deemed suitable for publication in PLOS ONE. Congratulations! Your manuscript is now being handed over to our production team.

Kind regards, 

on behalf of

Dr. Sona Frankova 

Academic Editor

PLOS ONE